# A Part Consolidation Design Method for Additive Manufacturing based on Product Disassembly Complexity

**Samyeon Kim [1] and Seung Ki Moon [2],***

[1] Digital Manufacturing and Design Centre, Singapore University of Technology and Design, Singapore 487372, Singapore; samyeon_kim@sutd.edu.sg

[2] Singapore Centre for 3D Printing, School of Mechanical & Aerospace Engineering, Nanyang Technological University, Singapore 639798, Singapore

* Correspondence: skmoon@ntu.edu.sg; Fax: +65-6792-4062

**Abstract:** Parts with complex geometry have been divided into multiple parts due to manufacturing constraints of conventional manufacturing. However, since additive manufacturing (AM) is able to fabricate 3D objects in a layer-by-layer manner, design for AM has been researched to explore AM design benefits and alleviate manufacturing constraints of AM. To explore more AM design benefits, part consolidation has been researched for consolidating multiple parts into fewer number of parts at the manufacturing stage of product lifecycle. However, these studies have been less considered product recovery and maintenance at end-of-life stage. Consolidated parts for the manufacturing stage would not be beneficial at end-of-life stage and lead to unnecessary waste of materials during maintenance. Therefore, in this research, a design method is proposed to consolidate parts for considering maintenance and product recovery at the end-of-life stage by extending a modular identification method. Single part complexity index (SCCI) is introduced to measure part and interface complexities simultaneously. Parts with high SCCI values are grouped into modules that are candidates for part consolidation. Then the product disassembly complexity (PDC) can be used to measure disassembly complexity of a product before and after part consolidation. A case study is performed to demonstrate the usefulness of the proposed design method. The proposed method contributes to guiding how to consolidate parts for enhancing product recovery.

**Keywords:** additive manufacturing; complexity; modular design; part consolidation; product recovery

## 1. Introduction

Studies of product design and development have helped engineers design products systematically. Product architecture has been determined to improve manufacturability of conventional manufacturing. A part with complex geometry in the product architecture divides into multiple parts for enhancing manufacturability due to limitations of conventional manufacturing. Accordingly, design for manufacturing and assembly (DFMA) has been focused on minimizing assembly and disassembly time and cost as well as managing complexity of products by minimizing the number of parts and connectors [1–3]. Since design freedom is severely restricted by conventional design methodologies, it is difficult to achieve optimal product architecture by consolidating parts [4,5].

Additive manufacturing (AM) is revolutionizing product development by fabricating parts with complex geometry directly [6]. Design for AM (DFAM) is introduced to improve manufacturability of AM and alleviate manufacturing constraints for AM, while product lifecycle and sustainability are

less considered. To explore design benefits by AM, part consolidation design methods have received attractions from designers in terms of product redesign for improving performance, but are still developing to integrate multiple parts, that are designed by limitations of conventional manufacturing, as a single part by applying AM capabilities. Accordingly, in this study, we propose a design method to consolidate parts for product recovery at the end-of-life (EOL) stage by extending conventional module identification process. Since a module consists of multiple parts, these parts in the identified module can be consolidated into a single object by AM. In the proposed method, product disassembly complexity (PDC) is used to measure difficulty while disassembling parts from a product. Therefore, the PDC plays an important role in understanding the status of product design for product recovery at the EOL stage. Since the PDC increases according to difficulty of disassembly of parts and the number of the parts and interfaces, the proposed design method aims to group parts with high disassembly difficulty into modules in order to minimize the disassembly complexity of the product at EOL stage. To assess disassembly difficulty in part level, single part complexity index (SCCI) is introduced by modifying the PDC to consider part and interface complexities simultaneously. Based on the SCCI, modules are identified by grouping parts with high SCCI value. The identified modules are considered as design boundary for part consolidation that can be fabricated by AM, so that they contribute to improving product recovery processes.

In this paper, Section 2 describes previous research and background in part consolidation and design for additive manufacturing, and then the proposed method is explained in Section 3. The proposed method described how to consolidate parts based on product disassembly complexity. Then a case study is performed with a coffee maker to demonstrate the usefulness of the proposed method in Section 4. A discussion of this study is described in Section 5. Closing remarks and future work are presented in Section 6.

## 2. Literature Review

Additive manufacturing (AM) process enables to produce complex parts. The AM has been evolved from rapid prototyping, which is to create a part or system rapidly as a prototype, to develop manufacturing process for creating final products directly. It alleviates design and manufacturing constraints, so that design freedom is extremely expanded [7]. In this sense, design for additive manufacturing (DFAM) has been introduced to take full advantage of the design freedom with concerning part consolidation and redesign, and hierarchical structures [6]. Most of previous studies in DFAM are to enhance performance of products while reducing costs [4,8,9], improve functional performance [10], and focus on design guidelines to print parts successfully under AM limitations [11]. Ponche, et al. [12] proposed a new DFAM methodology to consider design requirements and manufacturing specifications. The new DFAM methodology consists of three processes: part orientation and functional optimization for satisfying design requirements, and manufacturing paths optimization. Rosen [13] proposed a computer aided DFAM based on a process-structure-property-behavior framework to support part modeling, process planning, and manufacturing simulations. Thompson, et al. [4] explored design opportunities, benefits, and freedoms of AM at a part level and the macro scale, at the material level and the micro scale, and at a product level. They described part consolidation as a process to consolidate parts for assembly into a single printable object [14]. In other words, the part consolidation is considered to minimize the number of parts.

DFAM methodologies in previous studies focused on redesign of parts by using lattice structure and topology optimization. And, the redesign in module level and system level has been less addressed. According to AM capability, multi-parts can be merged as a single object instead of manufacturing and assembled parts separately and assembled. The advantages of the part consolidation are to improve manufacturing efficiency by avoiding assembly operations and reduce production cost by minimizing usage of connectors and tools for assembly [15]. There are few studies about the part consolidation. Liu [15] performed a comparative study to investigate improvement of structural performance through

the part consolidation. It results in a guideline that both structural topology and build direction should be optimized to improve structural performance of consolidated parts simultaneously. Becker, et al. [16] introduced design rules for AM to help designers rethink conventional assembly design towards part consolidation. Atzeni, et al. [17] also provided design rules for AM including part consolidation. The objective of the part consolidation was to redesign parts for conventional manufacturing and minimize production costs. However, these previous studies provided general design guidelines but had less focused on how to consolidate parts into a single object. Yang, et al. [18] proposed a method of consolidating parts for AM by considering function integration to achieve better functionality and structure optimization to improve performance at a part level. Moreover, when consolidating parts by AM, sustainability should be considered. Yang, et al. [19] proposed a framework to investigate environmental impact of consolidating parts on product lifecycle. It resulted in reduction of energy consumption and environmental impact when consolidating the parts by AM. In order to focus on the end-of-life stage of product lifecycle for sustainability, it needs to be considered product lifecycle and product recovery, especially maintenance, repair, and recovery when complex parts and products approach the end-of-life stage. The product recovery is a process of restoring inherent performance of retired products. By reusing the retired product and recycling materials, companies can minimize usage of raw materials, pollution during manufacturing, and wastes at the end-of-life stage [20,21]. In addition, by replacing obsolete parts to new parts, lifespan of products can be prolonged. Accordingly, when consolidating parts by using AM processes, the product recovery should be considered to improve sustainability. To facilitate product recovery, a disassembly process is necessary to detach materials, parts, and modules from the retired products.

The disassembly process can minimize cost and time for the product recovery, and avoid damage to the quality of detached parts [22]. Therefore, previous studies of design for disassembly is mainly focused on disassembly sequence planning [2,23,24]. As complete disassembly is not cost-effective and practical, the disassembly sequence planning emphasizes on selective disassembly for product recovery and maintenance. In some studies [25,26], attributes related to the difficulty of disassembly were considered and the disassembly sequences were decided based on disassembly cost. Regarding the importance of modular design for disassembly, Ishii, et al. [27] introduced module-based design for product retirement and evaluated the compatibility of modules by calculating disassembly time and cost. Kim and Moon [28] introduced a modular design method to generate eco-modules that consider disassembly efficiency, and reusability and recyclability. In terms of manufacturing process, it is needed to assess disassembly complexity for understanding current products' conditions and then planning design strategies based on the disassembly complexity. Several papers considered process complexity with design for assembly or disassembly. ElMaraghy and Urbanic [29] introduced a product and process complexity assessment tool to understand the effects of human workers' attributes in a manufacturing line. Samy and ElMaraghy [30] proposed a product assembly complexity tool with considering handling attributes and insertion attributes during assembly operation. These assessment tools for complexity would support assembly-oriented product design and guide designers to design products with less complexity. Soh, et al. [31] measured disassembly complexity based on design for assembly and accessibility for selective disassembly operations. Limitations of these researches are that interface complexity is less considered, although the interface complexity is a major aspect of disassembly operations. Therefore, this study emphasizes on an assessment of the product disassembly complexity based on interface and component complexities simultaneously.

From the literature, three issues are identified in terms of design guidelines and sustainability. First, the design guidelines and processes for part consolidation are less considered. Most of design guidelines emphasized only on reduction of the number of parts. Second, sustainability including product recovery has rarely been considered in design for additive manufacturing. Previous studies have been researched for improving functionality through redesign. However, there are no diverse reasons for part consolidation. Finally, to support the product recovery, it is required to understand and assess disassembly complexity of a product to identify parts with high disassembly difficulty

and facilitate disassembly operations. In the next section, the proposed part consolidation method to support AM is discussed in detail.

## 3. A Part Consolidation Design Method for Additive Manufacturing

Conventional modular design method aims to group multiple parts into modules to enhance manufacturing efficiency [32]. By shifting manufacturing paradigm from subtractive manufacturing to additive manufacturing, these multiple parts in a module can be considered as candidates for consolidation. Therefore, a part consolidation design method for AM, which is extending previous study [33,34], is proposed to group parts with high disassembly complexity into a module to enhance characteristics of products at the end-of-life (EOL) stage as shown in Figure 1. The first step is to understand function flows, such as material, signal, and energy flows, of products and physical relationships between parts. In the second step, single part complexity index (SCCI) is developed to provide information on which parts are difficult to disassemble for product recovery based on design attributes. The SCCI is an input of the third step and a modular driver for the product recovery to cluster modules from viewpoint of the EOL stage. In the third step, modules are identified based on adjacency matrix with the value of the SCCI by using Markov Cluster Algorithm. These modules would be assessed to check whether it can be manufactured by an AM technology in terms of material types. In this paper, since we focus on deciding clear design boundary for part consolidation regardless of manufacturing constraints of AM, material types are considered in this research. However, AM manufacturing constraints should be considered to determine more specific boundary for part consolidation after deciding specific AM processes. After that, parts in a module can be consolidated as a single object. It means that the concept of the module can be reinterpreted as the single part using the AM technology. Finally, to assess how product architecture with modules for part consolidation is improved to reflect product recovery, product disassembly complexity is used to compare between products with modules that is a set of parts and products with a consolidated part by AM.

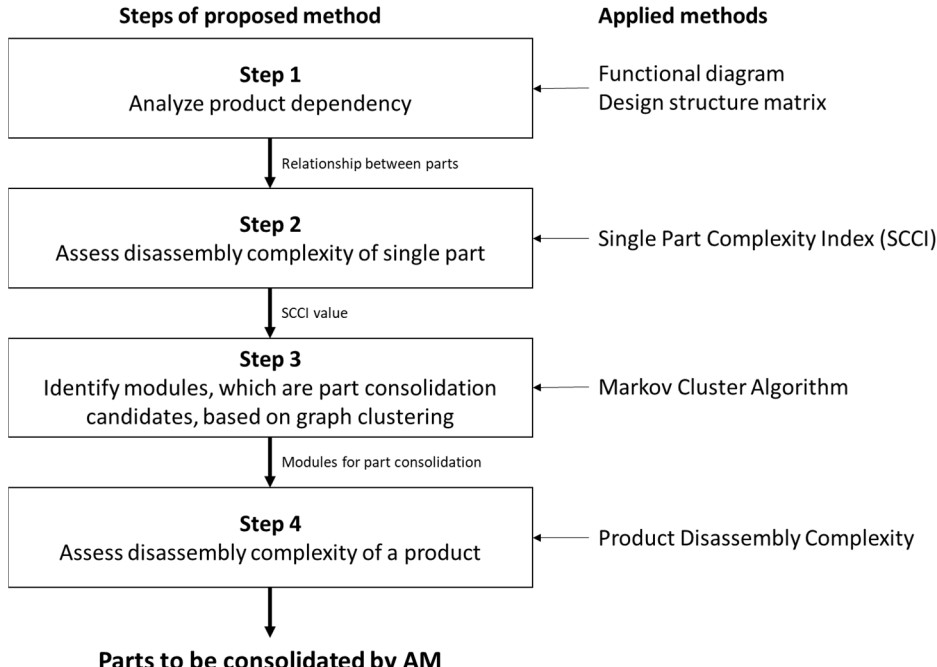

**Figure 1.** Overview of the proposed design method.

### 3.1. Product Dependency Analysis for Modular Design

Modular design has been developed to facilitate production processes, enhance product recovery including maintenance, and reduce the number of physical parts. The main principle of modular

design is to improve internal coupling within modules and minimize external coupling between modules [35]. Accordingly, when the main principle of modular design is extended to the field of additive manufacturing, it would be helpful to identify parts for consolidation. This is because module identification considers functional relationships, combinability, interface standardization, and interface complexity between parts [36]. Therefore, this paper mainly focuses on identifying modules that are candidates for part consolidation with considering product recovery. To identify modules, there are many tools for the modular design: axiomatic design, functional modeling, design structure matrix, and modular function deployment [36]. In this step, a functional diagram is used to understand the function flows of a product for identifying modules as shown in Figure 2. The functional diagram consists of boxed for describing functions and three function flows: energy, material, and signal flows. Based on this information, designers can classify modules heuristically like 'Heater' to 'Water reservoir' in Figure 2. A design structure matrix (DSM) tool is applied to determine relationships between parts in a product. As shown in Figure 3 of an example of DSM, '1' represents that two parts have a relationship, while '0' represents that there is no relationship. The DSM provides fundamental information to build an adjacent matrix in Step 3.

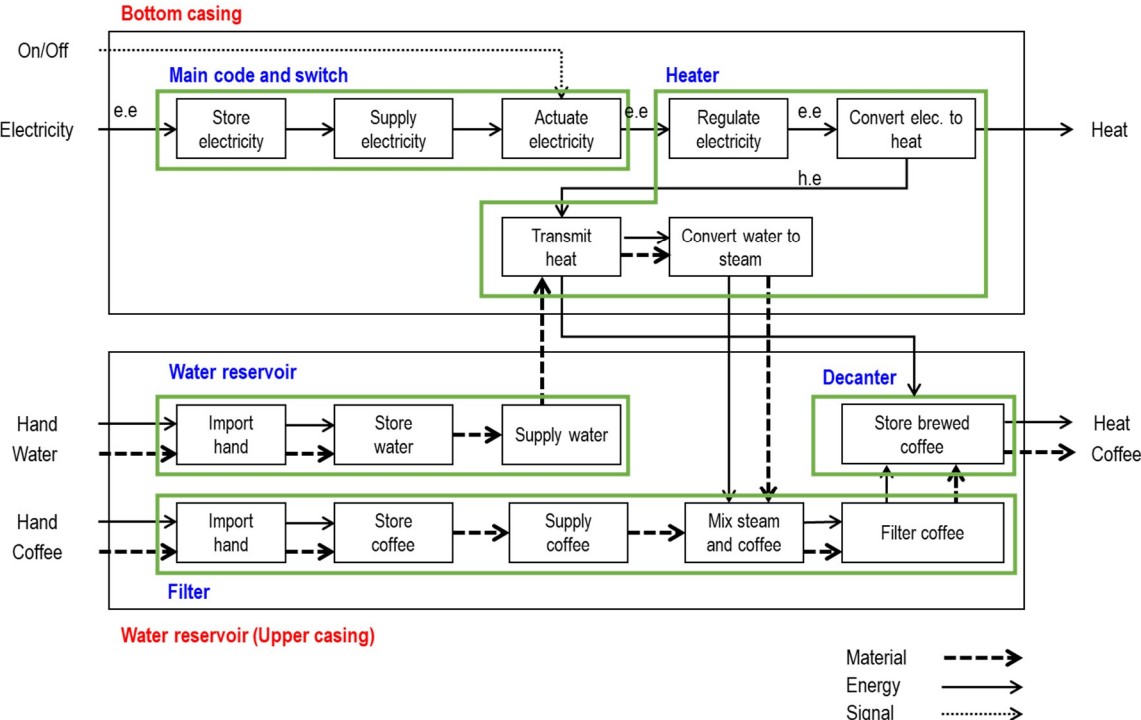

**Figure 2.** Functional diagram of the coffee maker.

|   | A | B | C | D | E |
|---|---|---|---|---|---|
| A |   | 1 | 1 |   |   |
| B | 1 |   | 1 |   |   |
| C | 1 | 1 |   | 1 |   |
| D |   |   | 1 |   | 1 |
| E |   |   |   | 1 |   |

**Figure 3.** An example of design structure matrix.

### 3.2. Assessment of Complexity of Single Part

This research considers a 'product disassembly complexity' term as the degree of disassembly difficulty [34]. The notion of the disassembly complexity has two levels: part complexity and interface complexity. For the part complexity, it emphasizes on attributes related to handling parts: weight effect factor, size, symmetry, and grasping parts. For the interface complexity, the connector, that links parts by physical and functional relationships, such as material, energy, and signal flows, is a key attributes for manual disassembly operations. The attributes for interface are related to mechanical connector types, non-mechanical connector types, and intensity of tool use. These attributes are critical to detach parts or modules from a product.

These attributes and corresponding descriptions for parts and interfaces are described in Table 1. These attributes are converted to the disassembly difficulty factor, which is values ranging from 0 to 1. The specific values of the disassembly factor are in reference [34]. Attributes that require high disassembly difficulty are close to 1, otherwise, 0. For the part complexity, values of the disassembly attributes for a part, called as disassembly difficulty factors, are determined by measuring assembly handling time and normalizing it based on [30]. For values of the interface complexity, U-rating values are applied to measure mechanical and non-mechanical unfastening processes. The U-rating value is developed by estimating disassembly efforts based on a survey by [37] and [38]. Since the range of the U-rating value is not between 0 and 1, the U-rating value is normalized in this study.

**Table 1.** Disassembly attributes for manual disassembly.

| Category | Attribute | Description |
|---|---|---|
| Part | Weight | This factor represents how difficult parts are positioned and handled according to part weight. Parts with heavy weight would need more man powers, extra tools like lift, and set-up time for parts and tools for disassembly. |
| | Size | A part size has an impact on both assembly and disassembly operations. When the component size is too small to grab it, it can delay the further disassembly process. |
| | Symmetry | The symmetry factor represents the easiness of disassembly process regarding directions for detaching parts and the difficulty of positioning parts for reassembly after disassembling the parts. |
| | Grasping and manipulation | Material property plays an important role in grasping parts, especially vulnerability and stiffness. Vulnerability entails damages or deformation of parts by dropping, bumping, and excessive grabbing force. Stiffness is the rigidity to resist deformation in response to an applied force, which is represented by elasticity modulus. |
| | | As a part with low vulnerability and high stiffness can be easily grasped by a worker, the disassembly difficulty factor' value will be low. Otherwise, the disassembly difficulty factor's value is closed to 1. |
| Interface | Mechanical unfastening process (U-rating) | As the mechanical connectors are detachable fasteners with relevant tools, it can be recursive for assembly and disassembly. In this research, nine types of the mechanical connectors are considered as follows: screw/bolt with standard head, screw/bolt special head, nut and bolt, retaining ring/circlips, interference fit, rivets/staples, pin, cylindrical snap fit, and cantilever snap fit. |
| | Non-mechanical unfastening (U-rating) | The non-mechanical connectors like lead and welding material are to firmly bond components, so that disassembly can be mostly difficult. |
| | Tools required with low intensity/ high intensity | When using the mechanical and non-mechanical connectors, relevant tools are needed for assembly and disassembly operations. The number of tools for disassembling parts and the intensity of the tool use are considered as a disassembly attribute to represent the difficulty of disassembly. |

By considering these disassembly attributes and their values, SCCI was introduced to analyze disassembly difficulty of a part by considering both part design and interface design at the same time as shown in the Equation (1) [39]. In Equation (1) for SCCI of the $k$th part, the weighted average value is applied to consider of part ($C_k$) and interface complexity indices ($I_k$).

$$SCCI_k = \frac{C_k \sum_1^J C_{c,j} + I_k \sum_1^N C_{i,n}}{\sum_1^J C_{c,j} + \sum_1^N C_{i,n}} \tag{1}$$

$$C_k = \frac{\sum_1^J C_{c,j}}{J} \tag{2}$$

$$I_k = \frac{\sum_1^N C_{i,n}}{N} \tag{3}$$

where, $C_{c,j}$ is a disassembly difficulty factor value of the $j$th attributes; $C_{i,n}$ is a disassembly difficulty factor value of $n$th interface attributes; $C_k$ is the average of disassembly difficulty factors for $k$th part; $J$ is the number of attributes for part complexity (here, $J = 4$); $I_k$ is the average of disassembly difficulty factors for interfaces of $k$th part; and $N$ is the number of attributes for interface complexity (here, $N = 3$) [39].

### 3.3. Module Identification based on Graph Clustering

In order to consider interwoven relationships between parts in a product, Markov Cluster Algorithm (MCL) is applied to group parts with high complexity into a module for AM. The MCL is used to cluster complex biological networks in the field of bioinformatics [40,41]. The MCL is a fast and scalable unsupervised clustering algorithm based on the mathematical concept of random walks.

First, an adjacent matrix, $A$, is developed with the value of the complexity as weight value on the edges. However, since the SCCI represents the disassembly complexity value of a single part, the SCCI value should be converted as the weight value of edges between $i$th part and $j$th part with the following equation.

$$A(i, j) = \begin{cases} w(i,j) & if\ ith\ and\ jth\ parts\ have\ relationships \\ 0 & else \end{cases} \tag{4}$$

$$w(i, j) = SCCI_i + SCCI_j \tag{5}$$

After building the adjacency matrix, second, Markov matrix, $M$, is developed to identify random walks from the adjacency matrix based on Equation (6). According to the equation, weight values in the adjacency matrix is transformed to values between 0 and 1 for representing stochastic flow from $i$th part to $j$th part.

$$M(i, j) = \frac{A(i, j)}{\sum_{k=1}^n A(k, j)} \tag{6}$$

Third, the MCL process performs two main operations: expansion and inflation. The expansion represents random walks with many steps and is the same as normal matrix multiplication. The expansion is to allow the flow to connect different regions of the graph. Nodes that have higher values with edges from a departure point to a destination point have high chance to be clustered. The inflation prunes edges with low disassembly complexity. By using Equation (7), the inflation operation makes regions with higher value on edges thicker, and makes regions with lower value on edges thinner based on the inflation parameter, $r$. The inflation parameter is non-negative value and used to rescale the matrix $M$. It results in $M_{inf}$, which is stochastic matrix and represents probability values of edges.

$$M_{inf}(i, j) = \frac{M(i, j)^r}{\sum_{k=1}^n M(k, j)^r} \tag{7}$$

By iterating these two main operations, parts will be grouped into modules, which is primary boundary of part consolidation for AM.

### 3.4. Assessment of Disassembly Complexity of a Product

Based on the aforementioned information in Table 1, the PDC can be used to represent a tendency of disassembly complexity of a product logarithmically. The total number of parts ($N_c$), the total number of interfaces ($N_i$), the number of unique parts ($n_c$), the number of unique interface ($n_i$), part complexity index (*CI*), and interface complexity index (*II*) are considered as the Equation (8) [34]. The PDC in Equation (8) is introduced by modifying the entropy theory. Accordingly, when the number of parts and interface, and values of CI and II are lower, the value of the PDC will be closed to 0.

$$PDC = \left(\frac{n_c}{N_c} + CI\right)log_2(N_c + 1) + \left(\frac{n_i}{N_i} + II\right)log_2(N_i + 1) \tag{8}$$

As shown in Equations (9) and (10), the *CI* and *II* are calculated to sum up part complexity and interface complexity of each part on Equations (2) and (3), respectively. The $w_k$ is a weight value of the interface complexity index.

$$CI = \sum_{1}^{n_p} w_k C_k \tag{9}$$

$$II = \sum_{1}^{n_p} w_k I_k \tag{10}$$

The PDC reflects design for disassembly that recommends reduction of the number of parts. When a product has less number of parts and interfaces, the PDC will be decreased. In this study, the PDC focuses on assessing part complexity and interface complexity for a product. PDC is used to assess disassembly complexity when a product consists of modules in conventional manufacturing or consolidated parts by AM processes.

### 3.5. Redesign for Additive Manufacturing

Parts are designed to alleviate manufacturing constraints of conventional manufacturing and enhance assembly efficiency to minimize manufacturing cost and time. Since design paradigm is shifting from conventional manufacturing to additive manufacturing, redesign for AM is required to alleviate newly introduced manufacturing constraints and add design values by AM. To utilize the advantages of AM technologies, designers must have understanding of AM capability and limitation to ensure manufacturability of parts because they do not have experience about AM and design for AM typically [42].

Consequently, existing design methods for conventional manufacturing have been modified and improved to consider AM. Two approaches are proposed to support the modification of existing design methods [42]: (1) a partial approach and (2) a global approach. The partial approach focuses on manufacturability improvement for AM so that the results are not very far from the conventional design. Since the partial approach starts with existing design but designers have a lack of DFAM knowledge, low AM design benefits can be taken. Filippi and Cristofolini [43] and Boyard, et al. [44] combined the Design for Manufacturing (DFM) and Design for Assembly (DFA), which are conventional design methods, to apply for DFAM. Filippi and Cristofolini [43] tried to build several knowledge matrices that combine the knowledge of both design-side and manufacturing-side. Boyard, et al. [44] developed a knowledge tree for AM that indicates the inter-connection between different design stages. On the other hands, the global approach is to support exploration of AM design benefits after selecting specific AM manufacturing process characteristics while meeting the functional requirements of the parts. Therefore, topology optimization method can be utilized to take advantages of AM by resolving the stress and strain distribution on a structure. The ultimate goal of topology is saving materials [9]. Yao,

Moon, and Bi (2017) proposed an AM design feature recommendation method that can help designers organize and utilize design knowledge to explore AM-enabled design space systematically. Both partial and global approaches can guide designers to redesign existing part for adopting AM by taking AM unique capabilities. Next, we demonstrate the effectiveness of the proposed design method using a case study involving a coffee maker.

## 4. Case Study

To demonstrate the usefulness of the proposed design method, a case study with a coffee maker was performed. The specification of the coffee maker is described in Table 2. In the first step, the function flows of the coffee maker were described to understand functional relationship between parts for identifying modules as shown in Figure 2. Then, DSM was developed to reflect the relationships between parts in the product as shown in Table 3. In the second step, each part design and interface design between parts in the product were analyzed by using Equations (2) and (3), respectively. Based on the analyzed values, SCCI is calculated by using Equation (1) as shown in Table 4. Each value of elements in the adjacency matrix was calculated by the sum of the SCCI values of two parts based on Equations (4) and (5), so that the adjacency matrix in Table 5 is determined finally. For example, a value of the element between bottom cover (1) and bottom casing (17) was 0.020 and it was calculated by the sum of SCCI value of the bottom cover, 0.010, and SCCI value of the bottom casing, 0.010.

In the third step, MCL was applied to determine modules for product recovery, which is a design boundary for part consolidation for AM as well, by using the adjacency matrix. Since MCL is an unsupervised learning algorithm, the number of modules is determined randomly. In this case study, the number of modules converges to 7 as shown in Table 6.

**Table 2.** Specification of the coffee maker.

| No. | Part Name | Material Type | Coffee Maker |
|-----|-----------|---------------|--------------|
| 1 | Bottom cover | PP | |
| 2 | Silicon ring | Silicon | |
| 3 | Hot plate | Al | |
| 4 | Casing for heater | PP | |
| 5 | Heater | Al | |
| 6 | Power cord | Copper | |
| 7 | Water tube set | PP | |
| 8 | Silicon tube | Silicon | |
| 9 | Water reservoir | PP | |
| 10 | Steam sprout | PP | |
| 11 | Filter basket | PP | |
| 12 | Filter frame | PP | |
| 13 | Filter net | PP | |
| 14 | Filter handle | PP | |
| 15 | Lid of coffee maker | PP | |
| 16 | Decanter | Glass | 4~6 cups/0.6 L |
| 17 | Bottom casing | PP | Brewing time <10 min |

**Table 3.** Design structure matrix of the coffee maker.

| DSM | | 1 | 2 | 3 | 4 | 5 | 6 | 7 | 8 | 9 | 10 | 11 | 12 | 13 | 14 | 15 | 16 | 17 |
|---|---|---|---|---|---|---|---|---|---|---|---|---|---|---|---|---|---|---|
| 1 | Bottom cover | | | | 1 | | | | | | | | | | | | | 1 |
| 2 | Silicon ring | | | 1 | 1 | | | | | | | | | | | | | |
| 3 | Hot plate | | 1 | | 1 | 1 | | | | | | | | | | | 1 | 1 |
| 4 | Casing for heater | 1 | 1 | 1 | | 1 | | | | | | | | | | | | 1 |
| 5 | Heater | | | 1 | 1 | | 1 | | 1 | | | | | | | | 1 | 1 |
| 6 | Power cord | | | | | 1 | | | | | | | | | | | | 1 |
| 7 | Water tube set | | | | | | | | 1 | | | | | | | | | |
| 8 | Silicon tube | | | | | 1 | | 1 | | 1 | 1 | | | | | | | |
| 9 | Water reservoir | | | | | | | | 1 | | | 1 | | | | 1 | | 1 |
| 10 | Steam sprout | | | | | | | | 1 | 1 | | 1 | | | | 1 | | |
| 11 | Filter basket | | | | | | | | | 1 | 1 | | 1 | | | 1 | | |
| 12 | Filter frame | | | | | | | | | | | 1 | | 1 | 1 | | | |
| 13 | Filter net | | | | | | | | | | | | 1 | | | | | |
| 14 | Filter handle | | | | | | | | | | | | 1 | | | | | |
| 15 | Lid of coffee maker | | | | | | | | | 1 | 1 | | | | | | | 1 |
| 16 | Decanter | | | 1 | | 1 | | | | | | 1 | | | | | | 1 |
| 17 | Bottom casing | 1 | | 1 | 1 | 1 | 1 | | | 1 | | | | | | 1 | 1 | |

**Table 4.** Complexity information of the coffee maker.

| No. | Part Name | No. | J | $C_k$ | N | $I_k$ | $SCCI_k$ |
|---|---|---|---|---|---|---|---|
| 1 | Bottom cover | 1 | 4 | 0.748 | 3 | 0.217 | 0.010 |
| 2 | Silicon ring | 1 | 4 | 0.828 | 3 | 0.327 | 0.013 |
| 3 | Hot plate | 1 | 4 | 0.748 | 3 | 0.217 | 0.010 |
| 4 | Casing for heater | 1 | 4 | 0.713 | 3 | 0.217 | 0.009 |
| 5 | Heater | 1 | 4 | 0.748 | 3 | 0.150 | 0.009 |
| 6 | Power cord | 1 | 4 | 0.748 | 3 | 0.483 | 0.014 |
| 7 | Water tube set | 1 | 4 | 0.748 | 3 | 0.150 | 0.009 |
| 8 | Silicon tube | 4 | 4 | 0.713 | 3 | 0.150 | 0.008 |
| 9 | Water reservoir | 1 | 4 | 0.788 | 3 | 0.110 | 0.010 |
| 10 | Steam sprout | 1 | 4 | 0.713 | 3 | 0.150 | 0.008 |
| 11 | Filter basket | 1 | 4 | 0.748 | 3 | 0.033 | 0.009 |
| 12 | Filter frame | 1 | 4 | 0.713 | 3 | 0.033 | 0.008 |
| 13 | Filter net | 1 | 4 | 0.788 | 3 | 0.277 | 0.011 |
| 14 | Filter handle | 1 | 4 | 0.748 | 3 | 0.110 | 0.009 |
| 15 | Lid of coffee maker | 1 | 4 | 0.788 | 3 | 0.133 | 0.010 |
| 16 | Decanter | 1 | 4 | 0.713 | 3 | 0.217 | 0.009 |
| 17 | Bottom casing | 1 | 4 | 0.748 | 3 | 0.217 | 0.010 |

In order to improve design feasibility of modules when adopting AM, manufacturing constraints of AM should be considered. Accordingly, total size of the module should be less than build chamber size of selected AM process and material types of parts in the module are identical except for using multi-material AM process. Furthermore, design rules for AM should be considered to improve manufacturability of product design. The design rules are mostly related to minimum thickness and overhang features that require support structure [45], which are derived from a combination of material and AM processes [46]. Therefore, designers should understand these various design rules.

In this study, we used the material type for assessing design feasibility of modules because the material type was critical when parts in a module were consolidated as a single part by sharing the same additive manufacturing processes. Accordingly, parts in modules 5 and 6 as shown in Figure 4 can be consolidated by using AM, which is 9' and 11' in Table 6. Accordingly, designers can consolidate parts in the modules 5 and 6 as a single part by using AM.

In the fourth step, the product disassembly complexity was applied to understand difficulty of disassembly and compare the difficulty of disassembly between a product with conventional modules and a product with consolidated parts in the modules 5 and 6. As a result, the product with consolidated

parts had a lower value of the PDC than the value of PDC of the product with conventional modules as shown in Table 7, which is around 19% PDC reduction by part consolidation.

**Table 5.** Adjacency matrix for the single part complexity index (SCCI) of the coffee maker.

| SCCI | | 1 | 2 | 3 | 4 | 5 | 6 | 7 | 8 | 9 | 10 | 11 | 12 | 13 | 14 | 15 | 16 | 17 |
|---|---|---|---|---|---|---|---|---|---|---|---|---|---|---|---|---|---|---|
| 1 | Bottom cover | | | | 0.019 | | | | | | | | | | | | | 0.020 |
| 2 | Silicon ring | | | 0.023 | 0.022 | | | | | | | | | | | | | |
| 3 | Hot plate | | 0.023 | | 0.019 | 0.019 | | | | | | | | | | | 0.019 | 0.020 |
| 4 | Casing for heater | 0.019 | 0.022 | 0.019 | | 0.018 | | | | | | | | | | | | 0.019 |
| 5 | Heater | | | 0.019 | 0.018 | | 0.023 | | 0.018 | | | | | | | | 0.018 | 0.019 |
| 6 | Power cord | | | | 0.023 | | | | | | | | | | | | | 0.023 |
| 7 | Water tube set | | | | | | | | 0.018 | | | | | | | | | |
| 8 | Silicon tube | | | | | 0.018 | | 0.018 | | 0.018 | 0.017 | | | | | | | |
| 9 | Water reservoir | | | | | | | | 0.018 | | 0.018 | 0.019 | | | | 0.020 | | 0.020 |
| 10 | Steam sprout | | | | | | | | 0.017 | 0.018 | | 0.017 | | | | 0.019 | | |
| 11 | Filter basket | | | | | | | | | 0.019 | 0.017 | | 0.017 | | | | 0.018 | |
| 12 | Filter frame | | | | | | | | | | | 0.017 | | 0.019 | 0.017 | | | |
| 13 | Filter net | | | | | | | | | | | | 0.019 | | | | | |
| 14 | Filter handle | | | | | | | | | | | | 0.017 | | | | | |
| 15 | Lid of coffee maker | | | | | | | | | 0.020 | 0.019 | | | | | | | 0.020 |
| 16 | Decanter | | | 0.019 | | 0.018 | | | | | | 0.018 | | | | | | 0.019 |
| 17 | Bottom casing | 0.020 | | 0.020 | 0.019 | 0.019 | 0.023 | | | 0.020 | | | | | | 0.020 | 0.019 | |

**Table 6.** Module identification and assessment.

| Module No. | A Product with Conventional Modules | A Product with Parts from AM | Assessment of Modules Material Type |
|---|---|---|---|
| 1 | 2, 3 | 2, 3 | X |
| 2 | 4 | 4 | - |
| 3 | 5 | 5 | - |
| 4 | 7, 8 | 7, 8 | X |
| 5 | 9, 10, 15 | 9' | O |
| 6 | 11, 12,13,14 | 11' | O |
| 7 | 1, 6, 16, 17 | 1, 6, 16, 17 | X |

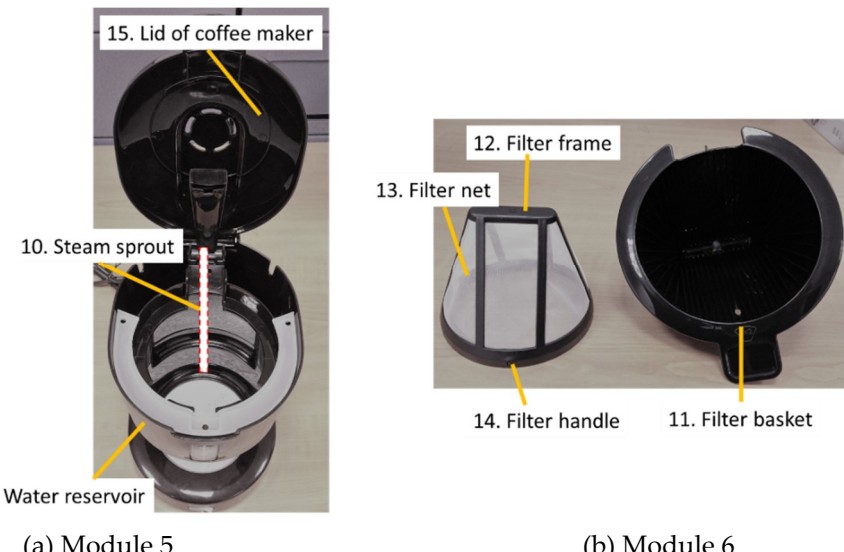

(a) Module 5                         (b) Module 6

**Figure 4.** Parts in selected modules for part consolidation.

**Table 7.** Comparison of product disassembly complexity (PDC) when considering modules and parts consolidation.

| Index | A Product with Conventional Modules | A Product with Parts from AM |
|---|---|---|
| $N_c$ | 20 | 15 |
| $n_c$ | 17 | 12 |
| $N_i$ | 8 | 8 |
| $n_i$ | 3 | 3 |
| PDC | 8.765 | 7.079 |

## 5. Discussion

Design for AM has mainly focused on creating parts with complex geometry for improving functionality, designing parts with considering constraints of AM processes, and consolidating parts for minimizing the number of parts. To consider product recovery including maintenance, part consolidation should be planned to achieve selective disassembly. Therefore, we proposed a design method to guide how to consolidate parts by removing assembly joints that are difficult to disassemble at the EOL stage. The proposed method results in modules based on the SCCI as a modular driver, and functional and physical relationships from a functional diagram and DSM.

After identifying these modules, it is required to check whether parts can be consolidated regarding material types of the parts. Since the parts in modules 1, 4, and 7 are made of different materials like aluminum, silicon, plastic, and glass, they cannot be consolidated due to limitations of AM processes that mostly support single material. On the other hands, modules 5 and 6 contain parts that have the same material and are closed to each other physically and functionally. Furthermore, since these parts are grouped into modules because they have high SCCI values, modules 5 and 6 are appropriate candidates for part consolidation to reduce the part count of a product, which is a primary goal of part consolidation. Modules 5 and 6 will be fabricated by AM, while other modules will be manufactured by conventional manufacturing. Accordingly, the result of the proposed design method can be used as design strategy to manage which parts will be fabricated by AM selectively to support flexible manufacturing by facilitating both conventional and additive manufacturing.

However, when designers consider a design feasibility factor as maintenance frequency of the parts instead of the material type between parts in the module, consolidating the filter basket and filter consisting of filter frame, filter net, and filter handle in module 6 may be not acceptable decision because the filter should be frequently cleaned after use. Furthermore, the proposed design method can be applied to generate new candidates for part consolidation, which are parts in modules, by considering other modular drivers related to repairability, reliability, or financial benefit. These modular drivers can be represented by characteristics of parts like SCCI and characteristics between parts. For example, remained useful lifespan (RUL) of each part can be modular drivers, and then parts with the same RUL can be grouped into a module by the proposed design method with using RUL of parts instead of SCCI. Since RUL of the parts is the same, maintenance frequency would be the same. Accordingly, parts with similar lifespan can be consolidated by AM. Furthermore, feasibility analysis for selected candidates for AM should be required to identify AM benefits in terms of redesign cost, manufacturing cost and time, financial benefit, and performance enhancement against subtractive manufacturing.

## 6. Closing Remarks and Future Work

AM enables fabricating parts with complex geometries and consolidating multiple parts for conventional manufacturing to enhance performance by using less material and energy, compared to subtractive manufacturing. However, design for AM has mainly focused on manufacturing stage in the product lifecycle rather than end-of-life (EOL) stage. Therefore, this study considers maintenance and product recovery at the EOL stage in order to prolong product lifecycle. Since disassembly operations are closely related to efficiency of reusability and recyclability in the EOL stage, we introduced the modular design method for consolidating multiple parts to less number of parts or a



single part. The disassembly complexity of each part is assessed by SCCI and then parts with high disassembly complexity are grouped into modules, which are candidates for part consolidation by using AM. Therefore, this study contributes to reduction of disassembly complexity of a product after the part consolidation.

A limitation of this study is to consider disassembly complexity for determining primary design boundary for part consolidation, which is the module. Accordingly, the proposed design method can be a starting point of product redesign for AM. As future work, other factors for product lifecycle, such as design cost, reliability of parts, maintenance requirements, and specific manufacturing constraints, will be considered to provide specific candidates for part consolidation within modules and between modules. After selecting these candidates, design feasibility of these candidates will be performed with various case studies with parts that have complex geometries after the part consolidation.

**Author Contributions:** Conceptualization, S.K. and S.K.M.; methodology, S.K.; formal analysis, S.K.; writing—original draft preparation, S.K.; writing—review and editing, S.K. and S.K.M.; supervision, S.K.M.; project administration, S.K.M.; funding acquisition, S.K.M. All authors have read and agreed to the published version of the manuscript.

**Funding:** This research was supported by the Singapore Centre for 3D Printing (SC3DP), the National Research Foundation, Prime Minister's Office, Singapore under its Medium-Sized Centre funding scheme.

**Conflicts of Interest:** The authors declare no conflict of interest.

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
