# Peer review of "A Part Consolidation Design Method for Additive Manufacturing based on Product Disassembly Complexity"

_applsci, doi:10.3390/app10031100_

Round 1
Reviewer 1 Report
COMMENTS AND SUGGESTIONS:
Comment 1: In the text, reference numbers should be placed in square brackets, for example [1], [1, 2, 3]. Now in the paper references are presented as: (Kim et al. 2015, Lambert 2003, Asikoglu and Simpson 2012).
Please, prepare references in the paper according to Template file and Instructions for authors.
Comment 2: Mathematical expressions is necessary to put in the center form in the text (see Template file).

Author Response
Reviewer 1
Comment 1: In the text, reference numbers should be placed in square brackets, for example [1], [1, 2, 3]. Now in the paper references are presented as: (Kim et al. 2015, Lambert 2003, Asikoglu and Simpson 2012). Please, prepare references in the paper according to Template file and Instructions for authors.
We have revised the reference format according to journal template in the manuscript.
Comment 2: Mathematical expressions is necessary to put in the center form in the text (see Template file).
We have updated all format of the manuscript based on the template including the mathematical expression position.Reviewer 2 Report
The premise of the paper is interesting and is a useful line of inquiry. However, the description in the abstract and introduction are rather different. The description in the introduction is better in my opinion. It appears that the abstract and the rest of the paper were written by different people and are inconsistent in several places.
Authors should address the sustainability of additive manufacturing more holistically. While material usage is higher, the energy used in make in the parts often increases significantly. The sustainability of AM often is questionable for many applications.
The terminology is unclear with the term “complexity” sometimes used where it appears that the authors are referring to “disassembly complexity”. At times there are references to the “complexities of the parts” which sounds as though it is about the geometry of the part rather than the complexity of its disassembly.
How are issues like maintenance requirements considered in this method?
It is unclear whether the method addresses the manufacturing limitations of the AM processes. This issue should be discussed.
The first paragraph of section 3.5 is confusing and unclear. It seems to say opposite things in different parts.
The case study is not very compelling in its outcomes. The module 6 would seem to be difficult to consolidate as the filter needs to be removed from the basket for periodic cleaning. Module 5 may be feasible but it is difficult to tell from the information provided.
This paper would be strengthened by additional examples.
Paper needs to be edited for English quality throughout. Subject verb agreement “when maintenance” – incomplete phrase “designed by limitation of conventional manufacturing” – unclear phrase “And, the AM has been” – multiple grammatical issues “has been evolved” Inconsistent verb tenses within a single sentenceI gave up listing them before the end of the 2nd page because there were just too many issues.
Author Response
Reviewer 2
1. The premise of the paper is interesting and is a useful line of inquiry. However, the description in the abstract and introduction are rather different. The description in the introduction is better in my opinion. It appears that the abstract and the rest of the paper were written by different people and are inconsistent in several places.
We have revised the abstract by aligning with the introduction. Also, we have revised other sections in this paper to improve readability and understanding.
2. Authors should address the sustainability of additive manufacturing more holistically. While material usage is higher, the energy used in make in the parts often increases significantly. The sustainability of AM often is questionable for many applications.
We agree with your opinion. AM is often questionable for many applications whether it is sustainable. Accordingly, we have deleted sentences that describe sustainability of AM technology to highlight how design for AM and design for disassembly make product sustainable.
3. The terminology is unclear with the term “complexity” sometimes used where it appears that the authors are referring to “disassembly complexity”. At times there are references to the “complexities of the parts” which sounds as though it is about the geometry of the part rather than the complexity of its disassembly.
We have updated all manuscripts based on your comments for usage of the term ‘complexity’. Accordingly, we didn’t use the complexity as single word, but use ‘disassembly complexity’ or parts with complex geometry’ to clarify the meaning of complexity.
4. How are issues like maintenance requirements considered in this method?
Maintenance requirements are not considered in this method, but this method has focused on how to improve disassembly operations for maintenance, recycle, and reuse. Considering maintenance requirements for part consolidation will be interesting future research. We mentioned it as future work in Section 6 on page 14.
5. It is unclear whether the method addresses the manufacturing limitations of the AM processes. This issue should be discussed.
We have addressed the manufacturing limitations of AM processes on page 12. Volume of build platform of AM machines and design rules about AM manufacturing constraints are discussed on page 12.
6. The first paragraph of section 3.5 is confusing and unclear. It seems to say opposite things in different parts.
We have revised the first paragraph of section 3.5.
7. The case study is not very compelling in its outcomes. The module 6 would seem to be difficult to consolidate as the filter needs to be removed from the basket for periodic cleaning. Module 5 may be feasible but it is difficult to tell from the information provided. This paper would be strengthened by additional examples.
Thank you for your suggestion. Since the proposed design method focuses on generating candidates for part consolidation, which is modules. Accordingly, we will further research about design feasibility within modules and any possibility of part consolidation between parts and between different modules as a case study. Furthermore, we will figure out more interesting case study with multiple examples for that. We have mentioned it as future work in Section 6 on page 14.
8. Paper needs to be edited for English quality throughout. Subject verb agreement “when maintenance” – incomplete phrase “designed by limitation of conventional manufacturing” – unclear phrase “And, the AM has been” – multiple grammatical issues “has been evolved” Inconsistent verb tenses within a single sentence. I gave up listing them before the end of the 2nd page because there were just too many issues.
We have proof-read all manuscript to avoid typos and grammatical errors.
Reviewer 3 Report
The subject of the reviewed article is interesting and important. However, the article itself does not bring new knowledge. The article is also not a scientific publication. The scientific goal has not been defined and the research gap which the author wants to fill has not been indicated
Author Response
Reviewer 3
1. The subject of the reviewed article is interesting and important. However, the article itself does not bring new knowledge. The article is also not a scientific publication. The scientific goal has not been defined and the research gap which the author wants to fill has not been indicated.
Thanks for your feedback. The purpose of this paper is to provide design methods to solve design problems by considering advanced manufacturing technology, e.g., AM. Our next goal is to solve design problems scientific way based on the proposed design method.
Reviewer 4 Report
Dear Authors,
the paper tackles an important aspect of additive manufacturing (AM), and should be considered for publication, if the Editor finds it considerably different from your previous publications.
However some further remarks are needed to be considered before full publication of the paper.
Abstract/Introduction/Literature survey:
- The abstract is not straightforward enough, the main definitions should be cleared in the introduction, not in the abstract
- Try to establish "Design for X" as "DfX (Design for X, where X stands for...)"
- Please try to establish literature connection to concepts or origin points for PDC or SCCI.
Discussion:
- Please refer to original source (or sources) of SCCI (1,2,3). Even if it was established/discussed in your previous works.
- Please omit commercialized brand naming in the paper. The coffee machine's number is enough for device identification.
- Is there a possibility to create such intricate construction as the discussed case study with AM? If so, what is the reality of mass production? What is the future prospect? Please evaluate and discuss!
- It is clear, that consolidation of parts can result in reduction of total part count, thus improving the disassembly. Also, multi-material AM is not really a viable solution. What aspects are on the other side of this simplified approach (financial, repairability, quality and realiability etc)? Please clarify in a deeper discussion!
- What are the further aspects of DfAM redesign for a product, which is in its lifecycle? How the redesign costs, tests, actual production costs would stand against subtractive solutions? Please emphasize!
Technical cleanliness, language, formatting, etc.:
- The paper is well written, however some polishing is recommended from the aspect of English use. (Also some minor style problems could be polished ... "The firstly/secondly", "difficult factors", "functional flows")
- The figures are clear and well written, but the generated abstract contains mostly pixel-based figures. Try to use as many vector based figures, as you can.
Author Response
Reviewer 4
Dear Authors,
The paper tackles an important aspect of additive manufacturing (AM), and should be considered for publication, if the Editor finds it considerably different from your previous publications. However, some further remarks are needed to be considered before full publication of the paper.
Abstract/Introduction/Literature survey:
- The abstract is not straightforward enough, the main definitions should be cleared in the introduction, not in the abstract
- Try to establish "Design for X" as "DfX (Design for X, where X stands for...)"
- Please try to establish literature connection to concepts or origin points for PDC or SCCI.
We have revised the abstract. We have deleted “Design for X” on page 1 and then use “Design for Manufacturing and Assembly” to avoid misunderstanding of the term “Design for X”. Also, we have updated literature review to link previous researches to concepts of PDC and SCCI on page 3.
Discussion:
- Please refer to original source (or sources) of SCCI (1,2,3). Even if it was established/discussed in your previous works.
We have added reference [39] for equations 1,2, and 3 for SCCI.
- Please omit commercialized brand naming in the paper. The coffee machine's number is enough for device identification.
We have removed the brand name of the product in the entire manuscript. Also, we have deleted Figure 7 on page 13 because of copyright issue. We believe that our description on page 12 would be good enough to make readers understand the result of the case study.
- Is there a possibility to create such intricate construction as the discussed case study with AM? If so, what is the reality of mass production? What is the future prospect? Please evaluate and discuss!
Creating parts with complex geometry is the goal of design for AM to explore design benefits. However, AM is not suitable for mass production, but for customization. Accordingly, design strategy is required to manage which parts will be fabricated by AM selectively, while other parts will be fabricated by conventional manufacturing for mass production. Therefore, this proposed design method will provide fundamental information on candidates for AM. Other modules which are not selected for AM can be fabricated by conventional manufacturing. Furthermore, the proposed design method highlights part consolidation by removing assembly joints which are difficult to disassemble at the end-of-life stage. We have discussed it on page 13.
- It is clear, that consolidation of parts can result in reduction of total part count, thus improving the disassembly. Also, multi-material AM is not really a viable solution. What aspects are on the other side of this simplified approach (financial, repairability, quality and realiability etc)? Please clarify in a deeper discussion!
Regarding other aspects for this proposed method and how these aspects are considered in the proposed method, we have added these contents on page 14 “ Furthermore, the proposed design method can be applied to generate new candidates for part consolidation, which are parts in modules, by considering modular drivers related to repairability, reliability, or financial benefit. These modular drivers can be represented by characteristics of parts like SCCI and characteristics between parts. For example, remained useful lifespan (RUL) of each part can be modular drivers, and then parts with same RUL can be grouped into a module by the proposed design method with using RUL of parts instead of SCCI. Since RUL of the parts is the same, maintenance frequency would be the same. Accordingly, parts with similar lifespan can be consolidated by AM.”
- What are the further aspects of DfAM redesign for a product, which is in its lifecycle? How the redesign costs, tests, actual production costs would stand against subtractive solutions? Please emphasize!
As the reviewer recommended, these additional aspects should be considered to determine concrete AM benefits against subtractive manufacturing. Therefore, we have added the contents on page 14 “Furthermore, feasibility analysis for selected candidates for AM should be required to identify AM benefits in terms of redesign cost, manufacturing cost and time, financial benefit, and performance enhancement against subtractive manufacturing.”
Technical cleanliness, language, formatting, etc.:
- The paper is well written, however some polishing is recommended from the aspect of English use. (Also some minor style problems could be polished ... "The firstly/secondly", "difficult factors", "functional flows")
We have proof-read to polish contents and avoid any grammatical errors.
- The figures are clear and well written, but the generated abstract contains mostly pixel-based figures. Try to use as many vector based figures, as you can.
We have converted figures with high resolutions.Round 2
Reviewer 3 Report
I have no further comment.